# Restoring Multidrug-Resistant *Escherichia coli* Sensitivity to Ampicillin in Combination with (−)-Epigallocatechin Gallate

**DOI:** 10.3390/antibiotics13121211

**Published:** 2024-12-13

**Authors:** Anong Kiddee, Atchariya Yosboonruang, Achiraya Siriphap, Grissana Pook-In, Chittakun Suwancharoen, Acharaporn Duangjai, Ratsada Praphasawat, Masami Suganuma, Anchalee Rawangkan

**Affiliations:** 1Division of Microbiology and Parasitology, School of Medical Sciences, University of Phayao, Phayao 56000, Thailand; anong.ki@up.ac.th (A.K.); atchariya.yo@up.ac.th (A.Y.); achiraya.si@up.ac.th (A.S.); grissana.po@up.ac.th (G.P.-I.); chittakun.su@up.ac.th (C.S.); 2Unit of Excellence on Research and Application of Natural Products for Health and Well-Being, University of Phayao, Phayao 56000, Thailand; acharaporn.du@up.ac.th (A.D.); ratsada.pr@up.ac.th (R.P.); 3Division of Physiology, School of Medical Sciences, University of Phayao, Phayao 56000, Thailand; 4Department of Pathology, School of Medicine, University of Phayao, Phayao 56000, Thailand; 5Graduate School of Science and Engineering, Saitama University, Saitama 338-8570, Japan; masami_suganuma@icloud.com

**Keywords:** antimicrobial resistant, ampicillin, EGCG, *Escherichia coli*, synergistic

## Abstract

Multidrug-resistant (MDR) bacteria, especially *Escherichia coli*, are a major contributor to healthcare-associated infections globally, posing significant treatment challenges. This study explores the efficacy of (−)-epigallocatechin gallate (EGCG), a natural constituent of green tea, in combination with ampicillin (AMP) to restore the effectiveness of AMP against 40 isolated MDR *E. coli* strains. Antimicrobial activity assays were conducted to determine the minimum inhibitory concentrations (MIC) of EGCG using the standard microdilution technique. Checkerboard assays were employed to assess the potential synergistic effects of EGCG combined with AMP. The pharmacodynamic effects of the combination were evaluated through time-kill assays. Outer membrane disruption was analyzed by measuring DNA and protein leakage and with assessments using N-phenyl-1-naphthylamine (NPN) and rhodamine 123 (Rh123) fluorescence dyes. Biofilm eradication studies involved biofilm formation assays and preformed biofilm biomass and viability assays. Scanning electron microscopy (SEM) was used to examine changes in cellular morphology. The results indicated that EGCG demonstrated activity against all isolates, with MICs ranging from 0.5 to 2 mg/mL, while AMP exhibited MIC values between 1.25 and 50 mg/mL. Importantly, the EGCG-AMP combination showed enhanced efficacy compared to either treatment alone, as indicated by a fractional inhibitory concentration index between 0.009 and 0.018. The most pronounced synergy was observed in 13 drug-resistant strains, where the MIC for EGCG dropped to 8 µg/mL (from 1 mg/mL alone) and that for AMP to 50 µg/mL (from 50 mg/mL alone), achieving a 125-fold and 1000-fold reduction, respectively. Time-kill assays revealed that the bactericidal effect of the EGCG-AMP combination occurred within 2 h. The mechanism of EGCG action includes the disruption of membrane permeability and biofilm eradication in a dose-dependent manner. SEM confirmed that the combination treatment consistently outperformed the individual treatments. This study underscores the potential of restoring AMP efficacy in combination with EGCG as a promising strategy for treating MDR *E. coli* infections.

## 1. Introduction

Multidrug-resistant (MDR) *Escherichia coli* has emerged as a significant threat to public health, along with MDR *Staphylococcus aureus*, *Klebsiella pneumoniae*, *Streptococcus pneumoniae*, *Acinetobacter baumannii*, and *Pseudomonas aeruginosa* [1]. The prevalence of MDR *E. coli* varies significantly across regions, with high resistance rates reported globally. In Europe, studies have shown that *E. coli* resistance to third-generation cephalosporins has reached concerning levels, with some reports indicating up to 40% of clinical isolates in certain countries exhibit resistance [2]. In the United States, the prevalence of carbapenem-resistant *E. coli* (CRE) has been rising, posing a serious challenge to treatment options [3]. Similarly, in parts of Asia and Africa, the spread of MDR *E. coli* is also a growing concern, with high rates of resistance observed in several countries [4]. The increasing prevalence of MDR *E. coli* is largely attributed to the overuse and misuse of antibiotics in both healthcare and agricultural settings, promoting the selection and spread of resistant strains [5]. In hospitals, MDR *E. coli* is a leading cause of healthcare-associated infections, often complicating the treatment of vulnerable patients [6]. Several new drugs, including ceftolozane/tazobactam, ceftazidime/avibactam, meropenem/vaborbactam, imipenem/cilastatin/relebactam, cefiderocol, plazomicin, eravacycline, and omadacycline, have recently been recommended for the treatment of Gram-negative multidrug-resistant bacterial infections [7,8]. However, ensuring their long-term effectiveness is crucial to delaying the emergence and spread of resistance to these novel agents. In response to the threat posed by MDR *E. coli*, novel therapeutic strategies are being explored, including the development of new antibiotics, and restoring sensitivity to existing drugs in combination therapies.

Ampicillin is a broad-spectrum beta-lactam antibiotic that has been widely used for a long time. The mechanisms of action of ampicillin are well known and involve interference with cell wall synthesis through attachment to penicillin-binding proteins (PBPs), inhibition of peptidoglycan synthesis, and inactivation of inhibitors of autolytic enzymes. Despite its clinical utility, the rising prevalence of antibiotic resistance has significantly reduced its effectiveness, particularly against Enterobacteriaceae like *E. coli*. Consequently, treatment with ampicillin alone is no longer recommended [9]. Sensitivity to ampicillin has been restored in the treatment of *Enterobacteriaceae*, including *E. coli*, through combination therapy with cloxacillin [10,11]. Moreover, combining ampicillin with ceftriaxone or azithromycin has demonstrated effectiveness in treating *Enterococcus faecalis* and *S. pneumoniae*, respectively [12,13]. However, attention should be given to the potential adverse effects, such as renal failure, although this is rare [14]. Additionally, these organisms are capable of preemptively adapting to antimicrobial resistance, leading to therapeutic failures. Consequently, combining ampicillin with natural products may provide viable alternative treatments for MDR *E. coli*.

Herbal medications are increasingly regarded as superior alternatives for addressing current and emerging antimicrobial-resistant bacteria and are expected to play a significant role in protecting humans against infections. Green tea (*Camellia sinensis*) is a rich natural source of polyphenols, including phenolic acids, such as caffeic acid and gallic acid, as well as flavonoids. Among these, catechins—a class of flavonoids containing flavan-3-ol units and galloylated catechins—are particularly prominent. Structurally, green tea catechins are characterized by a benzopyran framework with at least one aromatic ring [15]. Extensive research has highlighted the diverse health benefits of green tea consumption, including its antimicrobial properties against various pathogens. A 120 mL serving of green tea infusion contains approximately 150 mg of catechins, comprising 10–15% (−)-epigallocatechin gallate (EGCG), 6–10% (−)-epigallocatechin (EGC), 2–3% (−)-epicatechin gallate (ECG), and 2% (−)-epicatechin (EC) [16]. Among these catechins, EGCG, the most abundant and biologically active catechin in green tea, has gained attention in recent years not only for its antioxidant and anti-inflammatory properties but also for its potential to enhance the efficacy of anticancer, antiviral, and antibiotic therapies [16,17,18,19]. Upon consumption, these compounds and their metabolites are distributed throughout the body, aiding in both the treatment and prevention of infections. Green tea catechins exhibit antibacterial activity against both Gram-positive and Gram-negative bacteria through several mechanisms, including disruption of cell wall and membrane synthesis, inhibition of protein and nucleic acid synthesis, and interference with metabolic pathways involved in toxin production, extracellular matrix virulence factors, oxidative stress, and iron chelation [20]. Consequently, combining ampicillin with EGCG may offer a promising strategy to combat MDR *E. coli* antibiotic resistance and restore the efficacy of the conventional antibiotic treatment.

In this study, we investigated the efficacy of green tea EGCG against MDR *E. coli* strains, its combination with ampicillin, and the mechanisms underlying their antibacterial activity.

## 2. Results

### 2.1. EGCG Inhibits MDR E. coli Strains

We first investigated the susceptibility of 40 MDR *E. coli* strains to EGCG and ampicillin. Table 1 shows that all 40 isolates were sensitized to EGCG, with minimum inhibitory concentrations (MIC) ranging from 0.002 to 2 mg/mL and minimum bactericidal concentrations (MBC) ranging from 0.004 to 4 mg/mL. Notably, treatment with 1 mg/mL EGCG was effective against 87.5% (35/40) of the strains. In contrast, the MDR *E. coli* strains were resistant to ampicillin, with MIC values ranging from 1.25 to 50 mg/mL and MBC values ranging from 2.5 to 100 mg/mL. Interestingly, MDR *E. coli* strain E48, which was resistant to 13 antibiotics, including ampicillin (AMP), amoxicillin (AML), cephalothin (KF), cefotaxime (CTX), trimethoprim-sulfamethoxazole (SXT), meropenem (MEM), imipenem (IMP), gentamicin (CN), ciprofloxacin (CIP), norfloxacin (NOR), amoxicillin/clavulanic acid (AMC), ampicillin/sulbactam (SAM), and tetracycline (TE), had a MIC of 1 mg/mL for EGCG, while its MIC for AMP was 50 mg/mL. This suggests that EGCG is more effective at inhibiting MDR *E. coli* strains than ampicillin.

### 2.2. EGCG-AMP Combination Treatment Improved Antimicrobial Effectiveness Against MDR E. coli

The synergistic effects of EGCG, in combination with ampicillin against 40 MDR *E. coli* strains, were next investigated. As expected, the combination treatment produced a greater synergistic response compared to treatment with EGCG or ampicillin alone, as indicated by a fractional inhibitory concentration index (FICI) ranging from 0.009 to 0.018. This synergistic effect was observed in 92.5% (37/40) of the strains, as shown in Table 2. The combination treatment reduced the required EGCG dose by up to 125-fold and the ampicillin dose by up to 1250-fold. The most pronounced synergy was observed in the 13 drug-resistant strain, E48 (No. 34), where the MIC for EGCG dropped to 8 µg/mL (from 1 mg/mL alone) and the MIC for ampicillin decreased to 50 µg/mL (from 50 mg/mL alone), achieving 125-fold and 1000-fold reductions, respectively. These findings suggest that the EGCG-AMP combination could be a more effective treatment for MDR *E. coli* than either EGCG or ampicillin alone.

### 2.3. Analysis of the Bactericidal Kinetics of EGCG-AMP Combination Treatment

To explore the pharmacological activity of the EGCG-AMP combination treatment, we investigated the bactericidal effects on the kinetic growth curves of the representative MDR *E. coli* strain E48 under treatment with EGCG or ampicillin alone, or the EGCG-AMP combination. Figure 1 shows that treatment with 1 mg/mL of EGCG suppressed bacterial growth, exhibiting bactericidal activity within 18 h, while the control group displayed robust exponential growth, increasing to approximately 12 log units within 24 h. Interestingly, the EGCG-AMP combination treatment demonstrated bactericidal activity within 2 h, whereas ampicillin alone showed this effect within 4 h, indicating a 2-h reduction in the time to achieve bactericidal activity. These results clearly demonstrate that the combination of EGCG and ampicillin is bactericidal against MDR *E. coli*.

It is important to note that faster bactericidal kinetics are crucial in clinical settings to ensure rapid bacterial clearance, reduce infection duration, and minimize the likelihood of further resistance development.

### 2.4. EGCG-AMP Combination Treatment Disrupts the Membrane Permeability of MDR E. coli

Disruption of the cell membrane was next investigated by examining the leakage of nucleotides and proteins from bacterial cells following treatment with EGCG or ampicillin alone, or with their combination. The EGCG-AMP combination disrupts the membrane permeability of the MDR *E. coli* strain E48, as shown in Figure 2. After 1 h of treatment with various concentrations of EGCG (1, 2, and 4 mg/mL), DNA and protein leakage occurred in a dose-dependent manner. The EGCG-AMP combination treatment resulted in a higher rate of DNA release from the cells compared to treatments with either EGCG or ampicillin alone, demonstrating a strong effect on membrane permeability akin to that of Triton X-100 (Figure 2a). Similar effects were observed for protein leakage (Figure 2b). Additionally, the EGCG-AMP combination altered membrane permeabilization by increasing the relative fluorescence intensity (RFI) of NPN and decreasing the RFI of Rh123 more than treatments with EGCG or ampicillin alone (Figure 2c,d). Notably, EGCG exhibited a dose-dependent effect on outer membrane permeabilization. These results suggest that the EGCG-AMP combination modifies membrane potential and enhances membrane permeability, leading to the leakage of intracellular contents and subsequent cell death.

It is important to note that this disruption is a critical mechanism, as compromised membranes facilitate the entry of antibiotics like ampicillin, enhancing their bactericidal action. This finding underscores the potential of EGCG to act as a permeabilizer, aiding the antibiotic in bypassing traditional resistance barriers.

### 2.5. EGCG-AMP Combination Treatment Overcomes the Biofilm Recalcitrance of MDR E. coli

Biofilms are a key mechanism through which *E. coli* develops drug resistance. Therefore, the impact of the EGCG-AMP combination was explored on the disruption or elimination of biofilms. MDR *E. coli* E48 cells were subjected to the previously described treatment conditions. Figure 3a illustrates that EGCG effectively inhibited biofilm formation in a dose-dependent manner, achieving biomass inhibition rates of 75.7 ± 8.7%, 45.9 ± 12.4%, and 42.5 ± 11.9% at concentrations of 1, 2, and 4 mg/mL, respectively, compared to untreated cells. In contrast, treatment with 50 mg/mL of ampicillin resulted in only an 18.4 ± 4.3% reduction in biofilm biomass. Notably, the EGCG-AMP combination nearly eliminated biofilm biomass, resulting in a significant 93.3% reduction to 6.7 ± 1.5%.

The impact of the EGCG-AMP combination was also examined on the elimination of biofilms. One-day-old biofilms were treated with EGCG alone, ampicillin alone, or the combination for 24 h. Biofilm biomass was assessed using crystal violet staining and biofilm viability was evaluated through the MTT assay. Treatment with EGCG at concentrations of 1, 2, and 4 mg/mL resulted in decreases in biofilm biomass by 51.6 ± 24.3%, 31.5 ± 9.6%, and 29.4 ± 9.8%, respectively, compared to untreated biofilms (Figure 3b). The metabolic activity of the biofilms indicated that EGCG reduced the viability of MDR *E. coli* biofilm cells in a dose-dependent manner, with reductions of 71.38 ± 10.4%, 39.2 ± 9.3%, and 23.0 ± 5.9% at concentrations of 1, 2, and 4 mg/mL, respectively (Figure 3c). The EGCG-AMP combination was more effective at inhibiting both preformed biofilm biomass (3.92 ± 1.0% reduction) and biofilm viability (12.5 ± 1.8% reduction) than either treatment alone. These findings indicate that the EGCG-AMP combination can inhibit biofilm formation and eliminate existing biofilms of MDR *E. coli*, which is particularly relevant in treating chronic infections where biofilms play a critical role in persistence and resistance.

### 2.6. EGCG-AMP Combination Treatment Alters the Morphological Features of MDR E. coli and Impairs Biofilm Formation

To enhance our understanding of the effects of the EGCG-AMP combination treatment, scanning electron microscopy (SEM) was employed to examine the cellular morphology following a 4 h treatment (twice the bactericidal duration) with the individual agents and the EGCG-AMP combination. Figure 4 presents SEM images of *E. coli* E48 cells at 25,000× magnification. Untreated cells exhibited a smooth surface indicative of biofilm formation, characterized by an intact cell membrane and the absence of surface ruptures (Figure 4a). In contrast, treatment with EGCG alone resulted in minor membrane damage and a reduction in biofilm development (Figure 4b). Treatment with ampicillin alone led to significant membrane disruption and a substantial decrease in biofilm formation (Figure 4c). Notably, bacterial cells subjected to the EGCG-AMP combination exhibited pronounced membrane corrugations, cellular shrinkage, and rupture (Figure 4d). These findings indicate that the EGCG-AMP combination compromised cell membrane integrity and effectively eliminated biofilms, leading to morphological abnormalities that facilitated intracellular leakage, membrane retraction, and ultimately, cell death.

It is important to note that morphological changes, like cell wall deformation and shrinkage, suggest that the bacteria were stressed or damaged by the treatment. Impaired biofilm formation indicates that the combination disrupts the bacterial community’s ability to establish itself, which is crucial for preventing recurrent infections and biofilm-associated resistance.

## 3. Discussion

The WHO has highlighted MDR *E. coli* as a serious threat in efforts to combat antimicrobial resistance (AMR) [5]. This study suggests that combining EGCG with ampicillin could effectively overcome MDR *E. coli* infections by enhancing antibiotic efficacy, bypassing resistance mechanisms, and targeting resilient biofilms. This strategy exemplifies the restoration of antibiotic sensitivity, specifically in combination with natural compounds, as a novel approach against AMR.

Green tea is rich in catechins, a type of natural polyphenol, making up approximately 10–15% of its dry weight. The primary catechins in green tea include catechin, epicatechin, epigallocatechin, epicatechin gallate, and EGCG. Among these, EGCG is the predominant catechin, accounting for about 60% of the total, and has demonstrated the most potent antibacterial activity [21]. This aligns with our findings, which show that EGCG is effective in inhibiting all MDR *E. coli* strains. Furthermore, it is more effective than ampicillin, a drug that has largely fallen out of use. EGCG is well known for exhibiting multiple mechanisms of antibacterial action. It can disrupt bacterial cell membranes, increasing permeability and leading to the loss of essential cellular contents, which ultimately results in cell death [21]. Additionally, EGCG inhibits key bacterial enzymes, such as DNA gyrase and dihydrofolate reductase, which are crucial for DNA replication and cell survival [22]. EGCG also induces reactive oxygen species (ROS) formation by activating the Cpx system, leading to cell death [23]. Furthermore, EGCG induces oxidative stress within bacterial cells by generating ROS, which damages DNA, proteins, and lipids, leading to cell death [21,24,25]. EGCG also interferes with the formation of biofilms, protective barriers used by bacteria that evade antibiotics and immune responses, thereby reducing bacterial adhesion and biofilm integrity and rendering the bacteria more vulnerable to treatment [26,27]. Some studies suggest that EGCG may inhibit bacterial efflux pumps, mechanisms that bacteria use to expel antibiotics and other harmful compounds, thereby enhancing antibiotic retention and effectiveness within the cell [28,29].

This study found that the lowest EGCG concentration required to inhibit MDR *E. coli* was 2 µg/mL, as observed in strain E31. Conversely, the highest EGCG concentration needed for inhibition was 2 mg/mL, as observed in strains E40, E47, and E66. The most effective concentration across the tested strains was 1 mg/mL, which inhibited 35 out of 40 strains. Variations in susceptibility to EGCG among different MDR *E. coli* strains can be attributed to several factors, including differences in bacterial resistance mechanisms, such as efflux pumps, alterations in membrane permeability, or the presence of specific enzymes capable of degrading polyphenols like EGCG, as mentioned in the mechanisms above. Additionally, the genetic diversity of *E. coli* strains might play a significant role in their response to EGCG [30]. Strains exhibiting higher levels of intrinsic or acquired resistance mechanisms may require higher EGCG concentrations to achieve effective inhibition. Resistance to EGCG may also be linked to specific mutations in bacterial targets with which EGCG interacts, such as the bacterial cell wall or membrane-associated proteins. These mutations could result in reduced binding affinity for EGCG, thereby necessitating higher concentrations to achieve comparable inhibitory effects.

Notably, EGCG shows synergistic effects when used with certain antibiotics, enhancing their efficacy against drug-resistant bacteria, which is particularly useful against strains that have developed resistance mechanisms. Various studies indicate that combining EGCG with ampicillin enhances its effectiveness against methicillin-resistant *S. aureus* (MRSA) [31,32]. EGCG also demonstrates synergistic effects against *Vibrio cholerae* when used with tetracycline [21]. Additionally, combinations of EGCG with β-lactam antibiotics, such as carbapenem and meropenem, or carbenicillin, have been found to restore antibacterial activity against *A. baumannii* [33]. However, no synergistic effect had previously been reported between EGCG and ampicillin against *E. coli*. This study represents the first investigation of the combination of EGCG and ampicillin against MDR *E. coli*, demonstrating that the EGCG-AMP combination could be a more effective treatment for MDR *E. coli* than either EGCG or ampicillin alone.

A previous study reported that the combination of EGCG with gentamicin demonstrated a synergistic effect against MDR *E. coli*. Treatment with EGCG alone resulted in an MIC value of 1.25 mg/mL, while its combination with gentamicin reduced the MIC to 0.156 mg/mL, representing an 8-fold reduction. Similarly, gentamicin alone showed an MIC of 32 mg/mL, which decreased to 6.4 mg/mL when combined with EGCG, indicating a 5-fold reduction [34]. Therefore, in this study, the combination of EGCG with ampicillin may demonstrate even greater efficacy against MDR *E. coli* compared to gentamicin.

Although the MDR *E. coli* strain E31, which is resistant to six drugs (AMP, AML, KF, CTX, SXT, and SAM), exhibited good inhibitory activity with EGCG (2 µg/mL), the combination treatment of EGCG and ampicillin resulted in antagonism. This antagonism may be attributed to the strain’s high sensitivity to both agents. EGCG is known to interact with various biological molecules due to its phenolic structure, which enables it to bind to proteins, enzymes, and even antibiotics. This interaction could prevent these antibiotics from reaching their target sites or alter their functional efficacy, thereby demonstrating an antagonistic effect [35]. Consequently, we chose to focus on strain E48, which exhibits resistance to 13 drugs and demonstrates higher MIC values for EGCG (1 mg/mL) and ampicillin (50 mg/mL). The combination of EGCG and ampicillin in strain E48 produced clearer and more consistent results, enabling a significant reduction in the required concentrations of both agents (125-fold for EGCG and 1000-fold for ampicillin). This substantial improvement highlights strain E48 as a more suitable candidate for further investigation, as it better reflects the potential efficacy of the combination therapy in addressing high-level multidrug resistance.

Since we found that EGCG modifies membrane potential and enhances membrane permeability, leading to the leakage of intracellular contents, structural damage, and subsequent cell death, this represents a critical mechanism. Compromised membranes facilitate the entry of ampicillin, enhancing its bactericidal action and restoring ampicillin sensitivity. Therefore, this study supports the conclusion that bactericidal EGCG damages the cell membrane, decreasing its protective function, including protection from external stress [36,37].

Biofilm formation is a critical mechanism contributing to drug resistance in *E. coli*. Biofilms are complex, multicellular communities of bacteria encased in an extracellular polymeric substance (EPS) that confers resistance to antibiotics and environmental stresses [38]. Numerous studies have demonstrated that EGCG exerts significant effects on biofilm formation and disruption by interfering with the assembly of amyloid fibers and the production of phosphoethanolamine-modified cellulose fibrils [27,30,39]. Consistent with these findings, our study showed that EGCG reduced the viability of MDR *E. coli* biofilm cells in a dose-dependent manner and significantly decreased biofilm biomass. Notably, the combination of EGCG with ampicillin enhanced biofilm eradication more effectively than either treatment alone. These findings highlight the potential of EGCG as an adjunctive agent in addressing biofilm-associated resistance, offering a promising strategy for preventing recurrent infections and combating multidrug-resistant *E. coli*.

Together, these findings highlight EGCG as a promising candidate for use in antimicrobial therapies, especially in combination with traditional antibiotics to combat multidrug-resistant bacteria. Specifically, the combination treatment with ampicillin compromised cell membrane integrity and effectively eliminated biofilms, leading to morphological abnormalities that facilitated intracellular leakage, membrane retraction, and ultimately, cell death. Further studies could explore optimizing the dosage and delivery mechanisms for the EGCG-AMP combination to maximize its therapeutic efficacy and safety. Additionally, investigating the effectiveness of this combination against other MDR pathogens may broaden its clinical applicability.

## 4. Materials and Methods

### 4.1. Green Tea EGCG and Bacterial Strains

Epigallocatechin gallate (EGCG), a green tea catechin with a purity greater than 99% as determined by high-performance liquid chromatography (HPLC), was purified from Japanese green tea leaves (*Camellia sinensis* (L.) Kuntze, Theaceae) and prepared by the Saitama Prefectural Tea Institute, Saitama, Japan, as described in the previous report [40].

Multidrug-resistant (MDR) *E. coli* strains were isolated from houseflies at the Phayao Hospital in Phayao Province (Thailand), as previously reported [41]. These strains were tested against 15 antibiotics: chloramphenicol (C), amikacin (AK), AMP, AML, KF, CTX, SXT, MEM, IMP, CN, CIP, NOR, AMC, SAM, and TE. The resistance profiles for these MDR *E. coli* strains are presented in Appendix A. *E. coli* ATCC 25922 served as the reference strain.

### 4.2. Antimicrobial Activity Assay

The MIC of EGCG against MDR *E. coli* strains was assessed using the microdilution technique, following Clinical and Laboratory Standards Institute (CLSI) guidelines [42]. EGCG was serially diluted in Mueller-Hinton broth (HiMedia, Mumbai, Maharashtra, India) to obtain concentrations ranging from 1.95 µg/mL to 4.0 mg/mL. Bacterial cultures were prepared at a concentration of 5 × 10^5^ CFU/mL and incubated at 37 °C for 24 h. The MIC was determined using resazurin (ALPHA CHEMIKA, Mumbai, Maharashtra, India), as an indicator of bacterial viability. A solution of resazurin (1 mg/mL) was added at 10 μL per well, and the plates were further incubated for 4 h to observe any color change. Wells where the resazurin color remained blue, indicating no metabolic activity, were recorded as having concentrations above the MIC value [43]. The MBC was established based on the absence of colony growth on Mueller-Hinton agar (HiMedia, Mumbai, Maharashtra, India) in a drop test, as outlined in previous studies. Ampicillin (PanReac AppliChem, Barcelona, Spain), which is no longer recommended by the Centers for Disease Control and Prevention as a first-line treatment for *E. coli* infections, was also included in the testing.

### 4.3. Checkerboard Assays

Checkerboard assays were performed to analyze the potential synergistic effects of EGCG in combination with ampicillin, following previous reports [44,45]. The MDR *E. coli* E48 strain, which exhibited antibiotic resistance to up to 13 drugs, was used as the representative strain. The fractional inhibitory concentration index (FICI) was calculated using the following formulae:FIC (a) = MIC of EGCG in the combination/MIC of EGCG alone;(1)
FIC (b) = MIC of AMP in the combination/MIC of AMP alone;(2)
FICI = FIC (a) + FIC (b)(3)

A synergistic effect is indicated by an FICI of <0.5, an additive effect by an FICI of 0.5–4, and an antagonistic effect by a FICI > 4 [46,47].

### 4.4. Time-Kill Kinetics Assay

The pharmacokinetic effects of EGCG, in combination with ampicillin, were evaluated through a time-kill assay, as described in previous studies [48,49]. Concentrations of 1× MIC for EGCG alone, ampicillin alone, and the EGCG-AMP combination were tested to assess bacterial cell growth at intervals of 0, 1, 2, 4, 8, 16, and 24 h. The kill curve indicated a bactericidal effect with a reduction of ≥3 log_10_ CFU/mL compared to the initial inoculum (5 × 10^5^ CFU/mL), whereas a decrease of <3 log_10_ CFU/mL indicated a bacteriostatic effect.

### 4.5. Outer Membrane Disruption Analysis

The MDR *E. coli* E48 strain was evaluated for DNA and protein leakage from the cell membrane. EGCG at concentrations of 1×, 2×, and 4× MIC, 1× MIC ampicillin alone, and the EGCG-AMP combination were administered for 1 h at 37 °C. The amount of DNA released from the cytoplasm was measured at 260 nm using a NANO-400A Micro Spectrophotometer (Hangzhou Allsheng Instruments Co., Ltd., Hangzhou, China) to determine DNA concentration. Protein content was determined using a Bio-Rad DC Protein Assay Kit (Bio-Rad Laboratories, Inc., Hercules, CA, USA). Subsequently, the extent of outer membrane disruption was assessed through cell staining with N-phenyl-1-naphthylamine (NPN) (TCI, Tokyo, Japan) and rhodamine 123 (Rh123) fluorescence dyes (Sigma Aldrich, St. Louis, MO, USA) [50,51]. The fluorescence intensity of NPN (measured at OD 350/420 nm) and Rh123 (measured at OD 480/530 nm) was calculated using the formula:Relative fluorescence intensity (%) = [F1/F0] × 100,
where F0 represents the fluorescence intensity of untreated cells, and F1 denotes the fluorescence intensity of treated cells. A positive control consisting of 0.1% Triton X-100 (Sigma Aldrich, St. Louis, MO, USA) was also utilized.

### 4.6. Biofilm Formation Assay

The MDR *E. coli* E48 strain was inoculated at a concentration of 1 × 10^8^ CFU/mL in a 96-well microtiter plate, with a final volume of 200 μL, containing either EGCG alone (at 1×, 2×, and 4× MIC), ampicillin alone, or the EGCG-AMP combination. After 24 h of incubation, non-adherent cells were removed by gently washing with PBS, followed by heat-drying at 60 °C. Biofilm formation was assessed by staining with 200 μL of 0.1% crystal violet (ALPHA CHEMIKA, Mumbai, Maharashtra, India), for 15 min, followed by two washes with PBS. The crystal violet was then dissolved in 200 μL of 95% ethanol for 20 min at room temperature, and the optical density was measured at 595 nm. The percentage of biofilm mass was calculated using the formula:Biofilm mass (%) = [At/Ac] × 100, 
where Ac represents the OD595 for the control wells and At represents the OD595 in the presence of EGCG [45].

### 4.7. Preformed Biofilm Biomass and Viability Assay

Biofilms of the MDR *E. coli* E48 strain were grown for 24 h. Next, the planktonic cells were discarded, and the biofilm was rinsed with PBS. The appropriate treatment was then applied at a final volume of 200 μL and incubated for an additional 24 h at 37 °C. Biofilm biomass was evaluated using 0.1% crystal violet staining. The viability of the preformed biofilm was determined by incubating with 5 mg/mL of 3-(4,5-dimethylthiazol-2-yl)-2,5-diphenyltetrazolium bromide (MTT) (Sigma Aldrich, St. Louis, MO, USA) for 30 min at 37 °C. After the staining solution was discarded, dimethyl sulfoxide (DMSO) (Sigma Aldrich, St. Louis, MO, USA) was added to the wells and incubated for 30 min, after which the optical density was measured at 570 nm [45,52].

### 4.8. Scanning Electron Microscopy

The MDR *E. coli* E48 strain at a concentration of 1 × 10^8^ CFU/mL was treated with 1× MIC of both EGCG and ampicillin, as well as their combination, at 37 °C for 4 h. Following treatment, bacterial cells were collected by centrifugation at 3000× *g* for 5 min. The pellet was then resuspended in 1 mL of 2.5% glutaraldehyde (Sigma Aldrich, St. Louis, MO, USA) and incubated for 3 h at 4 °C. After washing with PBS, the bacterial cells underwent dehydration through a series of ethanol gradients (30%, 50%, 70%, and 90%) for 30 min each and were prepared for scanning electron microscopy (SEM; TESCAN, Vega III, Brno, Czech Republic), following established protocols [45].

### 4.9. Statistical Analysis

All experiments were conducted in at least triplicate. The data are expressed as mean ± standard deviation (SD). A one-way ANOVA followed by Dunnett’s multiple comparison test was performed using GraphPad Prism 5.01 (GraphPad Software, Inc., La Jolla, CA, USA). Statistical significance was defined as *p* < 0.05.

## 5. Conclusions

These findings highlight the significance of natural compounds like EGCG in restoring the efficacy of existing antibiotics and addressing the growing challenge of antimicrobial resistance. EGCG demonstrated dose-dependent effectiveness against MDR *E. coli* by disrupting membrane permeability and eradicating biofilms. A concentration of 1 mg/mL of EGCG was effective against 87.5% of the tested strains. Furthermore, EGCG acts as a potent synergist when combined with ampicillin, significantly enhancing its activity against MDR *E. coli*. This combination reduced the required dose of ampicillin by up to 1250-fold and EGCG by 125-fold, depending on the strain, offering a promising strategy to reintroduce ampicillin as a valuable tool in combating antimicrobial resistance.

Future research should focus on optimizing the dosage and delivery mechanisms of the EGCG-AMP combination and evaluating its efficacy against other MDR pathogens to maximize its therapeutic potential. Conducting clinical trials to assess the safety and efficacy of this combination therapy in human populations will be critical for translating these findings into viable treatment options for resistant infections. Additionally, the source of green tea and the extraction method of EGCG may significantly influence its antimicrobial activity and the outcomes of biological assays. As purified analytical-grade EGCG is relatively expensive, future experiments, including in vivo studies or clinical trials, should explore the use of commercially available EGCG as a cost-effective alternative. These insights contribute to the development of novel, natural compound-based strategies to counteract the global threat of multidrug-resistant pathogens.

## Figures and Tables

**Figure 1 antibiotics-13-01211-f001:**
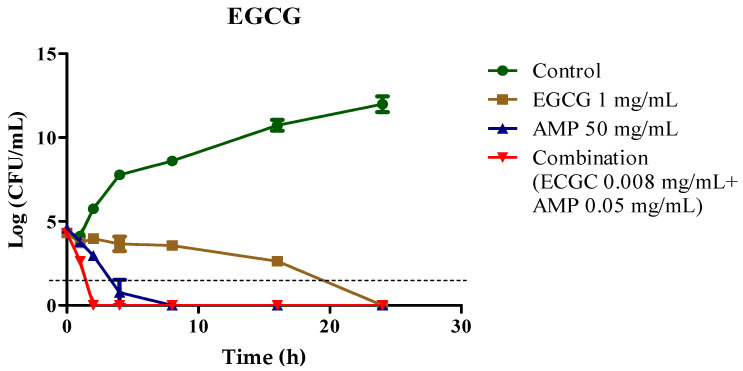
Effect of EGCG-AMP combination on MDR *E. coli* E48 strain. Time-kill kinetics for EGCG, ampicillin (AMP), and their combination were analyzed. Bacterial samples were taken at 1, 2, 4, 8, 16, and 24 h to measure viable cell counts. Dashed bars indicate the bactericidal threshold.

**Figure 2 antibiotics-13-01211-f002:**
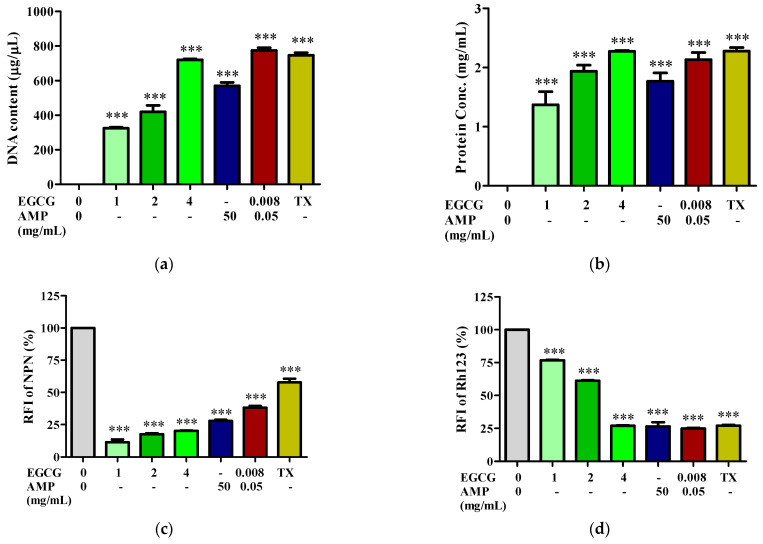
Effect of EGCG-AMP combination on membrane permeability. The *E. coli* E48 strain was exposed to EGCG alone (at 1×, 2×, and 4× MIC), AMP alone, or the EGCG-AMP combination for 1 h at 37 °C. DNA (**a**) and protein (**b**) levels were measured. The relative fluorescence intensity (RFI) of NPN (**c**) and Rh123 (**d**) was also assessed. Triton X-100 (0.1%) served as the positive control (TX). Significant differences compared to untreated controls are indicated by asterisks (*** *p* < 0.001).

**Figure 3 antibiotics-13-01211-f003:**
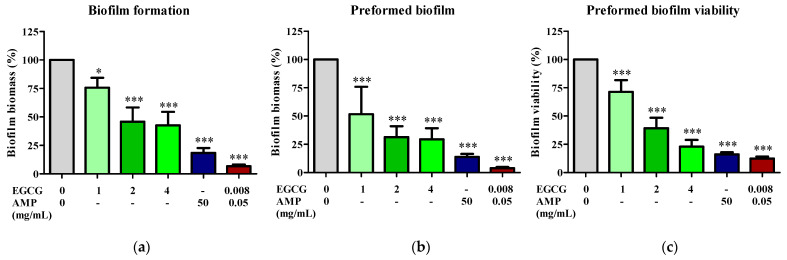
Effect of EGCG-AMP combination on biofilm formation. The *E. coli* E48 strain was exposed to EGCG alone (at 1×, 2×, and 4× MIC), AMP alone, or the EGCG-AMP combination for biofilm formation assays, as well as assays measuring preformed biofilm biomass and viability. Biofilm formation (**a**) and preformed biofilm (**b**) were evaluated using crystal violet staining. The viability of preformed biofilm (**c**) was assessed via the MTT assay. Significant differences compared to untreated controls are indicated by asterisks (* *p* < 0.05, *** *p* < 0.001).

**Figure 4 antibiotics-13-01211-f004:**
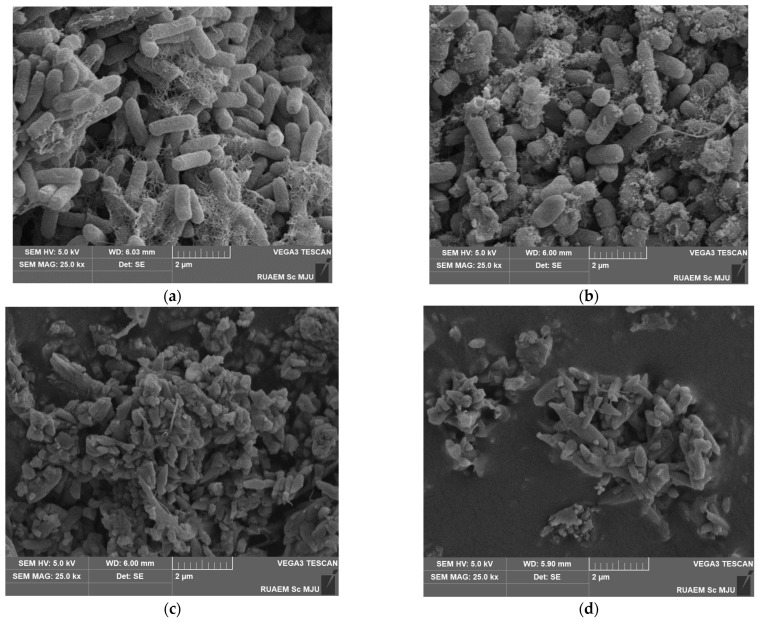
Effect of EGCG-AMP combination on bacterial cell morphology. The *E. coli* E48 strain was exposed for 4 h at 37 °C to either EGCG or AMP alone, and to their combination. Scanning electron microscopy (SEM) images at a magnification of 25,000× illustrate (**a**) the control; (**b**) EGCG at a concentration of 1 mg/mL; (**c**) AMP at a concentration of 50 mg/mL; and (**d**) treatment with the EGCG-AMP combination (EGCG 0.008 mg/mL + AMP 0.05 mg/mL).

**Table 1 antibiotics-13-01211-t001:** Susceptibility of 40 MDR *E. coli* strains to EGCG and ampicillin.

No	Isolates	EGCG (mg/mL)		AMP (mg/mL)	
		MIC	MBC	MIC	MBC
1	E1	1	2	1.25	2.5
2	E3	1	2	12.5	25
3	E5	1	2	5	10
4	E6	1	2	5	10
5	E8	1	2	5	10
6	E9	1	2	5	10
7	E10	1	2	3.125	6.25
8	E11	1	2	5	10
9	E14	1	2	5	10
10	E15	1	2	5	10
11	E16	1	2	5	10
12	E18	1	2	5	10
13	E19	1	2	5	10
14	E20	1	2	5	10
15	E21	1	2	1.25	2.5
16	E24	1	2	5	10
17	E25	1	2	2.5	5
18	E26	1	2	1.25	2.5
19	E27	1	2	2.5	5
20	E28	1	2	5	10
21	E29	1	2	2.5	5
22	E30	1	2	5	10
23	E31	0.002	0.004	5	10
24	E32	1	2	1.25	2.5
25	E34	1	2	2.5	5
26	E36	1	2	2.5	5
27	E37	1	2	2.5	5
28	E38	1	2	2.5	5
29	E39	1	2	2.5	5
30	E40	2	4	5	10
31	E41	1	2	12.5	25
32	E42	1	2	25	50
33	E47	2	4	5	10
34	E48	1	2	50	100
35	E49	1	2	5	10
36	E50	1	2	2.5	5
37	E52	1	2	1.25	2.5
38	E62	0.5	1	2.5	5
39	E65	1	2	5	10
40	E66	2	4	25	50
41	ATCC 25922	2	4	0.3125	0.625

Abbreviations: EGCG, (−)-epigallocatechin gallate; AMP, ampicillin; MIC, minimum inhibitory concentration; MBC, minimum bactericidal concentration.

**Table 2 antibiotics-13-01211-t002:** Synergistic effect of EGCG in combination with ampicillin against 40 isolates of MDR *E. coli*.

No.	Isolates	Alone MIC (mg/mL)		Combination MIC (mg/mL)		FIC [a]	FIC [b]	FICI	Outcome	Dose Reduction (Fold)	
		EGCG [a]	AMP [b]	EGCG [a]	AMP [b]					EGCG	AMP
1	E1	1	1.25	0.5	0.001	0.5	0.001	0.501	Additive	2	1250
2	E3	1	1.25	0.008	0.012	0.008	0.010	0.018	Synergy	125	104.2
3	E5	1	5	0.008	0.005	0.008	0.001	0.009	Synergy	125	1000
4	E6	1	5	0.008	0.005	0.008	0.001	0.009	Synergy	125	1000
5	E8	1	5	0.008	0.005	0.008	0.001	0.009	Synergy	125	1000
6	E9	1	5	0.008	0.005	0.008	0.001	0.009	Synergy	125	1000
7	E10	1	5	0.008	0.005	0.008	0.001	0.009	Synergy	125	1000
8	E11	1	5	0.008	0.005	0.008	0.001	0.009	Synergy	125	1000
9	E14	1	5	0.008	0.005	0.008	0.001	0.009	Synergy	125	1000
10	E15	1	5	0.008	0.005	0.008	0.001	0.009	Synergy	125	1000
11	E16	1	5	0.008	0.005	0.008	0.001	0.009	Synergy	125	1000
12	E18	1	5	0.008	0.005	0.008	0.001	0.009	Synergy	125	1000
13	E19	1	5	0.008	0.005	0.008	0.001	0.009	Synergy	125	1000
14	E20	1	5	0.008	0.005	0.008	0.001	0.009	Synergy	125	1000
15	E21	1	1.25	0.008	0.001	0.008	0.001	0.009	Synergy	125	1250
16	E24	1	5	0.008	0.005	0.008	0.001	0.009	Synergy	125	1000
17	E25	1	2.5	0.008	0.002	0.008	0.001	0.009	Synergy	125	1250
18	E26	1	1.25	0.008	0.001	0.008	0.001	0.009	Synergy	125	1250
19	E 27	1	2.5	0.008	0.002	0.008	0.001	0.009	Synergy	125	1250
20	E 28	1	5	0.008	0.005	0.008	0.001	0.009	Synergy	125	1000
21	E29	1	2.5	0.008	0.002	0.008	0.001	0.009	Synergy	125	1250
22	E30	1	5	0.008	0.005	0.008	0.001	0.009	Synergy	125	1000
23	E31	0.002	5	1.484	0.001	742.19	0.000	742.19	Antagonism	0	5000
24	E32	1	1.25	0.008	0.001	0.008	0.001	0.009	Synergy	125	1250
25	E34	1	2.5	0.008	0.002	0.008	0.001	0.009	Synergy	125	1250
26	E36	1	5	0.008	0.005	0.008	0.001	0.009	Synergy	125	1000
27	E37	1	2.5	0.008	0.002	0.008	0.001	0.009	Synergy	125	1250
28	E38	1	2.5	0.008	0.002	0.008	0.001	0.009	Synergy	125	1250
29	E39	1	2.5	0.008	0.002	0.008	0.001	0.009	Synergy	125	1250
30	E40	2	5	0.016	0.005	0.008	0.001	0.009	Synergy	125	1000
31	E41	1	12.5	0.008	0.012	0.008	0.001	0.009	Synergy	125	1041.7
32	E42	1	25	0.008	0.024	0.008	0.001	0.009	Synergy	125	1041.7
33	E47	2	5	2.000	5.0	1.000	1.000	2.000	Additive	1	1
34	E48	1	50	0.008	0.05	0.008	0.001	0.009	Synergy	125	1000
35	E49	1	5	0.008	0.005	0.008	0.001	0.009	Synergy	125	1000
36	E50	1	2.5	0.008	0.002	0.008	0.001	0.009	Synergy	125	1250
37	E52	1	1.25	0.008	0.001	0.008	0.001	0.009	Synergy	125	1250
38	E62	0.5	2.5	0.008	0.002	0.016	0.001	0.017	Synergy	62.5	1250
39	E65	1	5	0.008	0.005	0.008	0.001	0.009	Synergy	125	1000
40	E66	2	25	0.016	0.024	0.008	0.001	0.009	Synergy	125	1041.7
41	ATCC	2	0.3125	0.016	0.0003	0.008	0.001	0.009	Synergy	125	1041.7

Abbreviations: EGCG, (−)-epigallocatechin gallate; AMP, ampicillin; MIC, minimum inhibitory concentration; FIC, fractional inhibitory concentration; FICI, fractional inhibitory concentration index.

## Data Availability

The corresponding author will provide the data used in this study upon reasonable request.

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
