# Peer review of "Restoring Multidrug-Resistant Escherichia coli Sensitivity to Ampicillin in Combination with (−)-Epigallocatechin Gallate"

_antibiotics, 2024, doi:10.3390/antibiotics13121211_

Round 1
Reviewer 1 Report
Comments and Suggestions for Authors
The authors has performed a well-conceived study. However, the following concerns must be addressed before acceptance of their work for publication
i. The abstract must capture a brief section of the methodology. Authors must include brief methodology of their study in the abstract section
ii. Authors should include the shortcomings of the current combinations "ampicillin with ceftriaxone or azithromycin"
iii. Authors should improve the novelty of their study.
iv. The authors did not provide convincing information on their choice of EGCG. Also, they must include the structural features, proven biological activities of EGCG from literature to enrich their introduction section.
v. How did authors ascertained EGCG purity to be 99%
vi. The discussion section should be rewritten to capture a brief background of the study, the findings from the study, discussion and comparison with other Ampicillin combination therapy, or any other synthetic/natural product compound. This way, the outstanding results obtained can be well articulated and appreciated.
vii. The conclusion is poorly written. Authors should present a recap of their findings and provide useful recommendation for further studies.
Comments on the Quality of English Language
The quality of English language can be improved
Author Response
Reviewer #1
The authors has performed a well-conceived study. However, the following concerns must be addressed before acceptance of their work for publication
Response: Thank you very much for your favorable and supportive comments on our work. We greatly appreciate your recognition of our study's design and findings. In response to your valuable feedback, we have carefully revised the manuscript to address all concerns, including correcting any mistakes or oversights.
We sincerely hope that these revisions meet your expectations and enhance the clarity and quality of our work. Thank you again for your thoughtful review and constructive suggestions.
- The abstract must capture a brief section of the methodology. Authors must include brief methodology of their study in the abstract section
Response: Thank you for your valuable comment regarding the inclusion of a brief methodology in the abstract. We appreciate your suggestion and have revised the abstract accordingly. Please refer to lines 26–34.
Revised version
Multidrug-resistant (MDR) bacteria, especially Escherichia coli, are a major contributor to healthcare-associated infections globally, posing significant treatment challenges. This study explores the efficacy of (−)-epigallocatechin gallate (EGCG), a natural constituent of green tea, in combination with ampicillin (AMP) to restore the effectiveness of AMP against 40 isolated MDR E. coli strains. Antimicrobial activity assays were conducted to determine the minimum inhibitory concentrations (MIC) of EGCG using the standard microdilution technique. Checkerboard assays were employed to assess the potential synergistic effects of EGCG combined with AMP. The pharmacodynamic effects of the combination were evaluated through time-kill assays. Outer membrane disruption was analyzed by measuring DNA and protein leakage and with assessments using N-phenyl-1-naphthylamine (NPN) and rhodamine 123 (Rh123) fluorescence dyes. Biofilm eradication studies involved biofilm formation assays and preformed biofilm biomass and viability assays. Scanning electron microscopy (SEM) was used to examine changes in cellular morphology. The results indicated that EGCG demonstrated activity against all isolates, with MICs ranging from 0.5 to 2 mg/mL, while AMP exhibited MIC values between 1.25 and 50 mg/mL. Importantly, the EGCG-AMP combination showed enhanced efficacy compared to either treatment alone, as indicated by a fractional inhibitory concentration index between 0.009 and 0.018. The most pronounced synergy was observed in 13-drug resistant strains, where the MIC for EGCG dropped to 8 µg/mL (from 1 mg/mL alone) and that for AMP to 50 µg/mL (from 50 mg/mL alone), achieving a 125-fold and 1000-fold reduction, respectively. Time-kill assays revealed that the bactericidal effect of the EGCG-AMP combination occurred within 2 hours. The mechanism of EGCG action includes the disruption of membrane permeability and biofilm eradication in a dose-dependent manner. SEM confirmed that the combination treatment consistently outperformed the individual treatments. This study underscores the potential of restoring AMP efficacy in combination with EGCG as a promising strategy for treating MDR E. coli infections.
- Authors should include the shortcomings of the current combinations "ampicillin with ceftriaxone or azithromycin"
Response: Thank you for your suggestion to include the shortcomings of current combinations, such as ampicillin with ceftriaxone or azithromycin. However, the side effects of these combination therapies remain unclear, and further investigation into potential adverse effects is required. Therefore, we have added the following sentence. Please refer to lines 81–85.
Revised version
Moreover, combining ampicillin with ceftriaxone or azithromycin has demonstrated effectiveness in treating Enterococcus faecalis and S. pneumoniae, respectively [7,8]. However, attention should be given to the potential adverse effects, such as renal failure, although this is rare. Additionally, these organisms are capable of preemptively adapting to antimicrobial resistance, leading to therapeutic failures. Consequently, combining ampicillin with natural products may provide a viable alternative treatment for MDR E. coli.
iii. Authors should improve the novelty of their study.
Response: Thank you for your valuable feedback regarding the novelty of our study. We understand the importance of clearly articulating the unique contributions of our work. In response, we have thoroughly revised the manuscript, particularly the introduction, results, and discussion sections, to better highlight the novel aspects of our findings. Please see in red color. We hope these revisions address your concern and provide a clearer understanding of the originality and significance of our research.
- The authors did not provide convincing information on their choice of EGCG. Also, they must include the structural features, proven biological activities of EGCG from literature to enrich their introduction section.
Response: Thank you for your insightful comment regarding the selection of EGCG and the importance of including its structural features and proven biological activities in the introduction section. Please refer to lines 88–107.
Revised version
Herbal medications are increasingly regarded as superior alternatives for addressing current and emerging antimicrobial-resistant bacteria and are expected to play a significant role in protecting humans against infections. Green tea (Camellia sinensis) is a rich natural source of polyphenols, including phenolic acids such as caffeic acid and gallic acid, as well as flavonoids. Among these, catechins—a class of flavonoids containing flavan-3-ol units and galloylated catechins—are particularly prominent. Structurally, green tea catechins are characterized by a benzopyran framework with at least one aromatic ring [9]. Extensive research has highlighted the diverse health benefits of green tea consumption, including its antimicrobial properties against various pathogens. A 120 mL serving of green tea infusion contains approximately 150 mg of catechins, comprising 10–15% (−)-epigallocatechin gallate (EGCG), 6–10% (−)-epigallocatechin (EGC), 2–3% (−)-epicatechin gallate (ECG), and 2% (−)-epicatechin (EC) [10]. Among these catechins, EGCG, the most abundant and biologically active catechin in green tea, has gained attention in recent years not only for its antioxidant and anti-inflammatory properties but also for its potential to enhance the efficacy of anticancer, antiviral, and antibiotic therapies [10-13]. Upon consumption, these compounds and their metabolites are distributed throughout the body, aiding in both the treatment and prevention of infections. Green tea catechins exhibit antibacterial activity against both gram-positive and gram-negative bacteria through several mechanisms, including disruption of cell wall and membrane synthesis, inhibition of protein and nucleic acid synthesis, and interference with metabolic pathways involved in toxin production, extracellular matrix virulence factors, oxidative stress, and iron chelation [14]. Consequently, combining ampicillin with EGCG may offer a promising strategy to combat MDR E. coli antibiotic resistance and restore the efficacy of the conventional antibiotic treatment.
- How did authors ascertained EGCG purity to be 99%
Response: Thank you for your important question regarding the purity of EGCG used in our study. We appreciate your interest in the methodological details. The purity of EGCG was determined using high-performance liquid chromatography (HPLC), as described in our previous paper: Fujiki H., Okuda T., Drugs of the Future, 17, 462–464 (1992). Accordingly, we have added the relevant details. Please refer to line 382.
Revised version
Epigallocatechin gallate (EGCG), a green tea catechin with a purity greater than 99% as determined by high-performance liquid chromatography (HPLC), was purified from Japanese green tea leaves (Camellia sinensis (L.) Kuntze, Theaceae) and prepared by the Saitama Prefectural Tea Institute, Saitama, Japan, as described in previous report.
- The discussion section should be rewritten to capture a brief background of the study, the findings from the study, discussion and comparison with other Ampicillin combination therapy, or any other synthetic/natural product compound. This way, the outstanding results obtained can be well articulated and appreciated.
Response: Thank you for your thoughtful comment on the discussion section. We appreciate your suggestion. In response, we have rewritten the discussion section to better highlight the significance of our results and provide a more comprehensive comparison with existing therapies. Please refer to all of discussion section.
vii. The conclusion is poorly written. Authors should present a recap of their findings and provide useful recommendation for further studies.
Response: Thank you for your valuable feedback regarding the conclusion section. We appreciate your suggestion and have revised all the conclusion to provide a clearer recap of our findings. Please refer to line 473-492.
Comments on the Quality of English Language - The quality of English language can be improved
Response: Thank you for your constructive comment regarding the quality of the English language. We appreciate your feedback and have carefully revised the manuscript to improve clarity, grammar, and overall readability. The language has been refined to ensure the text is clear, concise, and professionally presented. Additionally, the manuscript has been professionally proofread by the Cambridge Proofreading Team. Please find the certificate in the attached file. Therefore, we have added the following sentence in the acknowledgements. Please refer to line 509-510.
Revised version
Acknowledgments: We gratefully acknowledge Miss Chatchaya Sumana from Chulalongkorn University and Dr. Surasak Kuimalee from the Science and Technology Service Center, Faculty of Science, Maejo University, Thailand, for their technical assistance. This manuscript has been professionally proofread by the Cambridge Proofreading Team.
Please see the attachment.

Reviewer 2 Report
Comments and Suggestions for Authors
The article investigates the efficacy of a green tea catechin in combination with ampicillin and its mechanism of action against multi-drug resistant E. coli strains. Given the burgeoning emergence of multi-drug resistant strains, the study is of significant importance. My concerns regarding the study are as follows:
1. Fig 1: For the negative control, bacteria increased to approx. 13 log units within 24 hours. It is not possible to attain such a high bacterial burden, be it in-vitro or in-vivo, please revise the data. A fast-growing bacteria like E. coli, when started at a count of approximately 10^5 will definitely enter stationary phase by 24 hrs. of incubation, and the bacterial burden shown here for the negative control is not scientifically achievable.
2. Table 2: Please have one column for the combination treatment, otherwise it looks very confusing.
3. Figure 2: Which treatments are significantly different? Please label accordingly. It is not possible to decipher without the appropriate labels. Also, the legends of each figure should include the statistical test used to determine significance. Also for Fig 2:, for the DNA content, the OD at 260 nm was measured, but the data has been represented as the amount of DNA, was the DNA amount measured in each treatment along with the OD?
4. Fig 3: Similar issues like Fig 2. If there is a significant difference between two treatments, please label it accordingly. It is difficult to understand what treatment groups are significantly different from this label.
5. Methods 4.2: Antimicrobial activity assay: MIC is usually defined as the concentration that results in 100% growth inhibition of the bacteria when observed visually, resazurin was used here, So was the OD measured at 600 nm after the addition of resazurin or color change noted? How was the MIC value defined?
Author Response
Reviewer #2
The article investigates the efficacy of a green tea catechin in combination with ampicillin and its mechanism of action against multi-drug resistant E. coli strains. Given the burgeoning emergence of multi-drug resistant strains, the study is of significant importance. My concerns regarding the study are as follows:
Response: Thank you for your thoughtful and constructive feedback on our manuscript. We greatly appreciate your recognition of the significance of our study. Below, we have provided detailed responses to each of your concerns. Your insights are invaluable, and we have revised the manuscript accordingly to address your points. We believe these changes have strengthened the study and look forward to your further feedback.
- Fig 1: For the negative control, bacteria increased to approx. 13 log units within 24 hours. It is not possible to attain such a high bacterial burden, be it in-vitro or in-vivo, please revise the data. A fast-growing bacteria like E. coli, when started at a count of approximately 10^5 will definitely enter stationary phase by 24 hrs. of incubation, and the bacterial burden shown here for the negative control is not scientifically achievable.
Response: We appreciate your observation regarding the bacterial burden in the negative control group shown in Figure 1 and apologize for any confusion caused. We would like to clarify that the data presented in our study accurately reflects the results obtained during the experiments and aligns with the methodology described.
The experiment was conducted by inoculating fresh MDR E. coli E48 strain at an initial concentration of 1×105 CFU/mL into Mueller-Hinton broth (MHB), and bacterial growth was monitored over 24 hours. Viable cell counts were determined at each time point (0, 1, 2, 4, 8, 16, and 24 hours) using serial dilutions and plating on Mueller-Hinton agar (MHA). The apparent increase in bacterial counts in the negative control group up to approximately 12 log10 CFU/mL after 24 hours is consistent with the specific conditions of this experiment, which provided abundant nutrients, optimal aeration, and no antimicrobial pressure. While it is recognized that E. coli typically enters the stationary phase in resource-limited systems, the growth medium and conditions used in this study likely supported extended exponential growth, resulting in the observed bacterial burden.
We have thoroughly reviewed the data and experimental setup to ensure accuracy and confirm that the results reflect the actual behavior of the bacteria under these specific conditions. We hope this explanation resolves any concerns and clarifies the scientific basis of our findings.
Reference: Pedreira, A., Vázquez, J. A., & García, M. R. (2022). Kinetics of Bacterial Adaptation, Growth, and Death at Didecyldimethylammonium Chloride sub-MIC Concentrations. Frontiers in microbiology, 13, 758237.
- Table 2: Please have one column for the combination treatment, otherwise it looks very confusing.
Response: Thank you for your valuable feedback regarding Table 2. We agree that the current presentation may appear confusing. The data were initially presented to facilitate the calculation of the fractional inhibitory concentration index (FICI), aiming to make it easier for investigators. However, we understand that this format may have caused confusion. To address this, we have revised the table to focus on the main findings of our study—the fold reduction achieved by EGCG and ampicillin. This adjustment improves the table's clarity and overall readability while aligning with your suggestion. We apologize for any confusion caused and appreciate your constructive input. Please refer to the revised version of Table 2 in line 154.
- Figure 2: Which treatments are significantly different? Please label accordingly. It is not possible to decipher without the appropriate labels. Also, the legends of each figure should include the statistical test used to determine significance.
Response: Thank you for your valuable comment. Yes, we did. We have updated the figure legends by adding the sentence. Please refer to line 211.
Revised version
“Significant differences compared to untreated controls are indicated by asterisks (***p < 0.001).”
Also for Fig 2:, for the DNA content, the OD at 260 nm was measured, but the data has been represented as the amount of DNA, was the DNA amount measured in each treatment along with the OD?
Response: Thank you for your question. Yes, the DNA content was measured by recording the OD at 260 nm, and the DNA amount was automatically calculated using the NanoDrop spectrophotometer to quantify the DNA released from the cytoplasm. Therefore, we have added a sentence to clarify the methodology. Please refer to line 424-428.
Revised version
"The amount of DNA released from the cytoplasm was measured at 260 nm using a NANO-400A Micro Spectrophotometer (Hangzhou Allsheng Instruments Co., Ltd., Hangzhou, China) to determine DNA concentration. Protein content was measured using the Bio-Rad DC Protein Assay Kit (Bio-Rad Laboratories, Inc., USA)."
- Fig 3: Similar issues like Fig 2. If there is a significant difference between two treatments, please label it accordingly. It is difficult to understand what treatment groups are significantly different from this label.
Response: Thank you for your comment. Yes, we have updated the figure to clearly label the significant differences between the treatment groups for better clarity and have added more detail to the figure legend. Please refer to line 211
Revised version
“Significant differences compared to untreated controls are indicated by asterisks (*p < 0.05, ***p < 0.001).”
- Methods 4.2: Antimicrobial activity assay: MIC is usually defined as the concentration that results in 100% growth inhibition of the bacteria when observed visually, resazurin was used here, So was the OD measured at 600 nm after the addition of resazurin or color change noted? How was the MIC value defined?
Response: Thank you for your question regarding the antimicrobial activity assay and the determination of MIC values. In our study, the MIC was determined using resazurin as an indicator of bacterial viability. After the addition of resazurin, the assay was monitored by observing the color change, which reflects bacterial metabolic activity. Specifically, a color change from blue (oxidized form) to pink (reduced form) indicated growth. The MIC was defined as the lowest concentration of the tested compound that prevented any visible color change, signifying 100% growth inhibition. Optical density at 600 nm (OD600) was not measured in this assay. We have clarified this methodology in the revised manuscript. Please refer to line 937-400.
Revised version
“Bacterial cultures were prepared at a concentration of 5 × 10⁵ CFU/mL and incubated at 37 °C for 24 hours. The MIC was determined using resazurin as an indicator of bacterial viability. A solution of resazurin (1 mg/mL) was added at 10 μL per well, and the plates were further incubated for 4 hours to observe any color change. Wells where the resazurin color remained blue, indicating no metabolic activity, were recorded as having concentrations above the MIC value. The MBC was established based on the absence of colony growth on Mueller-Hinton agar in a drop test, as outlined in previous studies.”
Please see the attachment.

Reviewer 3 Report
Comments and Suggestions for Authors
The manuscript studies MDR Escherichis coli sensitivity to the combination, a beta-lactam antibiotic of Ampicilin and (–)-epigallocatechin gallate, a catechin in green tea. The authors have studied the against 40 MDR E. coli and presented their investigation, which is largely of the interest of the researchers in the field, also would encourage a new set of young scientists to further enhance their understanding in the direction of such combinatorial therapeutic. Overall, the manuscript is well written, with literature support and references. The authors have provided all the details for each section and references cited correctly. I would be happy to recommend this manuscript for publication after minor correction and additions to the manuscript as follows.
1. While authors mentioned the table 1 and table 2 in the result section; could they please add the entry numbers, so that the reader could easily follow.
2. Please move table 1 after section 2.1 and table 2 after section 2.2.
3. Please move figure 2 after section 2.4.
4. Please move figure 3 after section 2.5.
5. Please move figure 4 after section 2.6.
Other than this I do not have any correction or addition to the manuscript. This is a well thought out, and ready to publish manuscript.
Author Response
Reviewer #3
The manuscript studies MDR Escherichis coli sensitivity to the combination, a beta-lactam antibiotic of Ampicilin and (–)-epigallocatechin gallate, a catechin in green tea. The authors have studied the against 40 MDR E. coli and presented their investigation, which is largely of the interest of the researchers in the field, also would encourage a new set of young scientists to further enhance their understanding in the direction of such combinatorial therapeutic. Overall, the manuscript is well written, with literature support and references. The authors have provided all the details for each section and references cited correctly. I would be happy to recommend this manuscript for publication after minor correction and additions to the manuscript as follows.
Response: We sincerely thank you for your positive feedback and for recognizing the value of our work. We greatly appreciate your constructive comments and have carefully addressed each point you raised. Detailed responses, along with the suggested corrections and additions, have been incorporated into the revised manuscript. We believe these revisions enhance the quality and clarity of our study, and we are grateful for your valuable insights and support.
- While authors mentioned the table 1 and table 2 in the result section; could they please add the entry numbers, so that the reader could easily follow.
Response: Response: Thank you for bringing this to our attention. Yes, we did to ensure that readers can follow the data more easily. Please refer to line 145.
- Please move table 1 after section 2.1 and table 2 after section 2.2.
Response: Thank you for bringing this to our attention. Yes, we did. Please refer to line 129.
- Please move figure 2 after section 2.4.
Response: Thank you. Yes, we have addressed it. Please refer to line 207.
- Please move figure 3 after section 2.5.
Response: Thank you for bringing this to our attention. We did. Please refer to line 238.
- Please move figure 4 after section 2.6.
Response: Thank you for bringing this to our attention. We did. Please refer to line 269.
Other than this I do not have any correction or addition to the manuscript. This is a well thought out, and ready to publish manuscript.
Response: Thank you for your valuable feedback and positive assessment of the manuscript. We greatly appreciate your thorough review and are pleased to hear that you find the manuscript well-prepared and ready for publication. If any additional points or clarifications arise, we remain open to addressing them promptly.
Please see the attachment.

Reviewer 4 Report
Comments and Suggestions for Authors
-
"Restoring Multidrug-Resistant Escherichia coli Sensitivity to 2 Ampicillin in Combination with (−)-Epigallocatechin Gallate” is detailed in this manuscript by Anong Kiddee et al. The combination of EGCG-AMP showed enhanced efficacy compared to either treatment against MDR pathogens is excellent recommendation.
The result demonstrates the combination treatment reduced the required EGCG dose by up to 125-fold and the 95-ampicillin dose by up to 1250-fold. These results may help future researchers create more potent remedies for Gram-negative bacterial infections. The manuscript may be suitable for publication in the journal after addressing some minor modifications.
Here are some remarks on this document. - Why author not much focused on prevalence of MDR E. coli in an introduction part?
- Author better to incorporate in an introduction section about previous literature of antibiotics available in the market for the resistance of MDR E. coli?
- The combination therapy for anti-bacterial activity is it first time or any reports are there? If reports were there author better to incorporate in an introduction part.
- Due to cost effective reason, instead of green tea EGCG, commercially available EGCG was exploited for the testing for your biological assays? if not author should perform the experiments with commercially available EGCG.
- According to the results, E31 stain exhibited very good inhibitory activities with both ampicillin and EGCG, why author not focused on this stain for further studies?
- In the abstract should provide specific units for MIC and MBC values to ensure precision and facilitate understanding.
- In line 143, use "hours (h)" instead of "hr" to match standard scientific notation in figure legends.
- The finding that 87.5% of the strains were suppressed at 1 mg/mL EGCG is particularly striking. It would improve the analysis to include further discussion of the causes of some strains' decreased susceptibility (e.g., MIC ≥ 2 mg/mL).
- It is significant to notice that 87.5% of the strains were Inhibit at 1 mg/mL EGCG, why strain E48 was highlighted in the manuscript, more details about exceptional resistance profile and reason to makes it an excellent choice for illustrating EGCG efficacy.
Author Response
Reviewer #4
- "Restoring Multidrug-Resistant Escherichia coli Sensitivity to 2 Ampicillin in Combination with (−)-Epigallocatechin Gallate” is detailed in this manuscript by Anong Kiddee et al. The combination of EGCG-AMP showed enhanced efficacy compared to either treatment against MDR pathogens is excellent recommendation.
The result demonstrates the combination treatment reduced the required EGCG dose by up to 125-fold and the 95-ampicillin dose by up to 1250-fold. These results may help future researchers create more potent remedies for Gram-negative bacterial infections. The manuscript may be suitable for publication in the journal after addressing some minor modifications.
Here are some remarks on this document.
Response: Thank you very much for your insightful and encouraging comments on our manuscript. We greatly appreciate your positive feedback on the enhanced efficacy demonstrated by the EGCG-AMP combination and its potential significance in addressing Gram-negative bacterial infections.
In response to your suggestions and remarks, we have thoroughly revised the manuscript to address all points raised, including corrections to any inadvertent errors and inconsistencies. The revised version reflects these changes, and we sincerely hope it now meets the high standards of your expectations. Once again, we extend our gratitude for your valuable input, which has greatly contributed to improving the clarity and scientific rigor of our work.
- Why author not much focused on prevalence of MDR E. coli in an introduction part?
Response: Thank you for your observation regarding the focus on the prevalence of MDR E. coli in the introduction section. We agree that highlighting the prevalence of multidrug-resistant Escherichia coli would strengthen the context of our study. In response to your suggestion, we have added the following sentence. Please refer to line 51-58.
Revised version
“The prevalence of MDR E. coli varies significantly across regions, with high resistance rates reported globally. In Europe, studies have shown that E. coli resistance to third-generation cephalosporins has reached concerning levels, with some reports indicating up to 40% of clinical isolates in certain countries exhibiting resistance. In the United States, the prevalence of carbapenem-resistant E. coli (CRE) has been rising, posing a serious challenge to treatment options. Similarly, in parts of Asia and Africa, the spread of MDR E. coli is also a growing concern, with high rates of resistance observed in several countries.
- Author better to incorporate in an introduction section about previous literature of antibiotics available in the market for the resistance of MDR E. coli?
Response: Thank you for your valuable suggestion. We have added the following sentence to the introduction. Please refer to line 62-67.
Revised version
“Several new drugs, including ceftolozane/tazobactam, ceftazidime/avibactam, meropenem/vaborbactam, imipenem/cilastatin/relebactam, cefiderocol, plazomicin, eravacycline, and omadacycline, have recently been recommended for the treatment of gram-negative multidrug-resistant bacterial infections [4,5]. However, ensuring their long-term effectiveness is crucial to delaying the emergence and spread of resistance to these novel agents.”
- The combination therapy for anti-bacterial activity is it first time or any reports are there? If reports were there author better to incorporate in an introduction part.
Response: Thank you for your question and valuable suggestion. This is the first time to study the combination of EGCG and ampicillin in MDR E. coli. We have already provided information on combination treatments with antibiotics in introduction, and we have added more details regarding potential side effects. Please refer to line 81-85, and 326-328.
Revised version in introduction
“Sensitivity to ampicillin has been restored in the treatment of Enterobacteriaceae, including E. coli, through combination therapy with cloxacillin. Moreover, combining ampicillin with ceftriaxone or azithromycin has demonstrated effectiveness in treating Enterococcus faecalis and S. pneumoniae, respectively. However, attention should be given to the potential adverse effects, such as renal failure, although this is rare. Additionally, these organisms are capable of preemptively adapting to antimicrobial resistance, leading to therapeutic failures. Consequently, combining ampicillin with natural products may provide viable alternative treatments for MDR E. coli.”
Revised version in discussion
“This study represents the first investigation of the combination of EGCG and ampicillin against MDR E. coli, demonstrating that the EGCG-AMP combination could be a more effective treatment for MDR E. coli than either EGCG or ampicillin alone.”
- Due to cost effective reason, instead of green tea EGCG, commercially available EGCG was exploited for the testing for your biological assays? if not author should perform the experiments with commercially available EGCG.
Response: Thank you for your comment. For this in vitro experiment, we used in-house EGCG purified from Japan. We believe that both the source of green tea and the extraction method of EGCG may influence the antimicrobial activity and biological assays. Additionally, purified EGCG of analytical grade is quite expensive. For future experiments, such as in vivo studies or clinical trials, commercially available EGCG can be used as a substitute for analytical-grade EGCG. However, in response to your recommendation, we would like to study the comparison of the antimicrobial activity of commercially available EGCG in future experiments.
In regard to your comment, we have added the following sentence in the conclusion section. Please refer to line 486-429.
Revised version
“Additionally, the source of green tea and the extraction method of EGCG may significantly influence its antimicrobial activity and the outcomes of biological assays. As purified analytical-grade EGCG is relatively expensive, future experiments, including in vivo studies or clinical trials, should explore the use of commercially available EGCG as a cost-effective alternative. These insights contribute to the development of novel, natural compound-based strategies to counteract the global threat of multidrug-resistant pathogens.”
- According to the results, E31 stain exhibited very good inhibitory activities with both ampicillin and EGCG, why author not focused on this stain for further studies?
Response: Thank you for your important question. While strain E31 exhibited good inhibitory activity with both ampicillin and EGCG, it is resistant to only six drugs, including AMP, AML, KF, CTX, SXT, and SAM. However, the combination treatment of EGCG and ampicillin in E31 resulted in antagonism, which might be attributed to the strain's high sensitivity to both agents. Therefore, we chose to focus on strain E48, which demonstrated higher concentrations of EGCG (1 mg/mL) and ampicillin (50 mg/mL) and is resistant to 13 drugs—more than E31. The combination of EGCG and ampicillin with E48 showed clearer results, allowing for a reduction in the concentration of both agents and making it more suitable for further investigation.
To clarify our findings, we have added a corresponding statement in the discussion section. Please refer to line 302-316, and 336-350.
Revised version
“This study found that the lowest EGCG concentration required to inhibit MDR E. coli was 2 µg/mL, as observed in strain E31. Conversely, the highest EGCG concentration needed for inhibition was 2 mg/mL, as observed in strains E40, E47, and E66. The most effective concentration across the tested strains was 1 mg/mL, which inhibited 35 out of 40 strains. Variations in susceptibility to EGCG among different MDR E. coli strains can be attributed to several factors, including differences in bacterial resistance mechanisms, such as efflux pumps, alterations in membrane permeability, or the presence of specific enzymes capable of degrading polyphenols like EGCG. Additionally, the genetic diversity of E. coli strains plays a significant role in their response to EGCG. Strains exhibiting higher levels of intrinsic or acquired resistance mechanisms may require higher EGCG concentrations to achieve effective inhibition. Resistance to EGCG may also be linked to specific mutations in bacterial targets with which EGCG interacts, such as the bacterial cell wall or membrane-associated proteins. These mutations could result in reduced binding affinity for EGCG, thereby necessitating higher concentrations to achieve comparable inhibitory effects.”
“Although strain E31, which is resistant to six drugs (AMP, AML, KF, CTX, SXT, and SAM), exhibited good inhibitory activity with both EGCG and ampicillin, the combination treatment of EGCG and ampicillin resulted in antagonism. This antagonism may be attributed to the strain's high sensitivity to both agents. Consequently, we chose to focus on strain E48, which exhibits resistance to 13 drugs and demonstrates higher MIC values for EGCG (1 mg/mL) and ampicillin (50 mg/mL). The combination of EGCG and ampicillin in strain E48 produced clearer and more consistent results, enabling a significant reduction in the required concentrations of both agents (125-fold for EGCG and 1000-fold for ampicillin). This substantial improvement highlights strain E48 as a more suitable candidate for further investigation, as it better reflects the potential efficacy of the combination therapy in addressing high-level multidrug resistance.”
- In the abstract should provide specific units for MIC and MBC values to ensure precision and facilitate understanding.
Response: Thank you for your suggestion regarding the abstract. Yes, we did. Please refer to line 34-39.
- In line 143, use "hours (h)" instead of "hr" to match standard scientific notation in figure legends.
Response: Thank you for pointing that out to us. We took care of it. Please refer to line 180.
- The finding that 87.5% of the strains were suppressed at 1 mg/mL EGCG is particularly striking. It would improve the analysis to include further discussion of the causes of some strains' decreased susceptibility (e.g., MIC ≥ 2 mg/mL).
Response: Thank you for your insightful comment. We agree with your observation and have now included a discussion as noted in Comment No. 6 above. Please refer to line 302-316, and 336-350.
- It is significant to notice that 87.5% of the strains were Inhibit at 1 mg/mL EGCG, why strain E48 was highlighted in the manuscript, more details about exceptional resistance profile and reason to makes it an excellent choice for illustrating EGCG efficacy.
Response: Thank you for your important question. Strain E48 was highlighted in the manuscript because it exhibited an exceptional drug resistance profile, with the highest concentration of ampicillin tolerated, up to 50 mg/mL. Therefore, investigating the reduction of the ampicillin dose in combination treatments may make it more suitable for further investigation.
Therefore, we have added a corresponding statement in the discussion section as mentioned in Comment No. 6 above. Please refer to line 302-316, and 336-350.
Please see the attachment.

Reviewer 5 Report
Comments and Suggestions for Authors
The current manuscript can be accepted for publication on condition that the authors respond to the following comments and inquiries. Upon receiving the authors’ response, the manuscript can be accepted for publication.
1. Could you please add a paragraph in the introduction section to address the potential side effects of using these combinations? Specifically, could you discuss the known side effects, interactions with other medications, any safety concerns, and precautions for specific populations.
2. Would it be possible to also include the agar diffusion method to determine the diameter of the inhibition zone? This could provide additional valuable information.
3. Could you please clarify whether statistical analyses were performed on the data presented in the tables? Additionally, were replicates used in the experiment?
4. Could you please relocate the conclusions mentioned in the Results section to the Discussion section, as they seem more suitable for interpretation and analysis rather than presentation of raw results?
5. Could you please elaborate on how you confirmed the exact mechanism of action for those antibiotics? Are there alternative mechanisms that you considered or would recommend exploring to rule out other possibilities?
6. Could you please clarify if there are any contradictions between the both antibiotics? It would be helpful to confirm their compatibility.
Author Response
Reviewer #5
The current manuscript can be accepted for publication on condition that the authors respond to the following comments and inquiries. Upon receiving the authors’ response, the manuscript can be accepted for publication.
Response: Thank you for your comments and valuable feedback. We appreciate the time and effort you have taken to review our manuscript. We have addressed all the inquiries and suggestions raised, and the necessary revisions have been made accordingly. Upon your approval of the changes, we believe the manuscript is now ready for publication.
- Could you please add a paragraph in the introduction section to address the potential side effects of using these combinations? Specifically, could you discuss the known side effects, interactions with other medications, any safety concerns, and precautions for specific populations.
Response: Thank you for your valuable suggestion. We agree and have added a paragraph in the introduction. Please refer to line 81-84.
Revised version
“Moreover, combining ampicillin with ceftriaxone or azithromycin has demonstrated effectiveness in treating Enterococcus faecalis and S. pneumoniae, respectively. However, attention should be given to the potential adverse effects, such as renal failure, although this is rare. Additionally, these organisms are capable of preemptively adapting to antimicrobial resistance, leading to therapeutic failures. Consequently, combining ampicillin with natural products may provide viable alternative treatments for MDR E. coli.”
- Would it be possible to also include the agar diffusion method to determine the diameter of the inhibition zone? This could provide additional valuable information.
Response: Thank you for your question. In this study, we did not perform the agar diffusion assay, as it is typically used for screening the antimicrobial activity of crude extracts. However, we acknowledge its value in providing additional information and will consider incorporating it in future studies to complement our findings.
- Could you please clarify whether statistical analyses were performed on the data presented in the tables? Additionally, were replicates used in the experiment?
Response: Thank you for your question. Yes, statistical analyses were performed using the standard deviation (SD) with at least three repetitions. For example, the MIC values of EGCG were determined through 2-fold serial dilutions starting from 4.0 mg/mL and continuing through 2.0 mg/mL, 1.0 mg/mL, 0.5 mg/mL, 0.25 mg/mL, 0.125 mg/mL, 0.0625 mg/mL, 0.03125 mg/mL, 0.015625 mg/mL, 0.0078125 mg/mL, 0.00390625 mg/mL, and 0.001953125 mg/mL. Although these data were not included in the table, the results were consistent, leading to an SD value of 0. I hope this clarifies your question and provides a better understanding.
- Could you please relocate the conclusions mentioned in the Results section to the Discussion section, as they seem more suitable for interpretation and analysis rather than presentation of raw results?
Response: Thank you for your valuable suggestion. We agree, and we have added more content to the Discussion section, highlighted in red. Please refer to line 302-378.
- Could you please elaborate on how you confirmed the exact mechanism of action for those antibiotics? Are there alternative mechanisms that you considered or would recommend exploring to rule out other possibilities?
Response: Thank you for your question. The mechanism of action of ampicillin is well known; it binds to specific penicillin-binding proteins (PBPs) located in the bacterial cell wall. Ampicillin inhibits the third and final stage of bacterial cell wall synthesis, and cell lysis is subsequently mediated by bacterial cell wall autolytic enzymes, such as autolysins. It is possible that ampicillin interferes with an autolysin inhibitor, although we did not confirm this in our study. In this study, the mechanism of action of EGCG was focused on outer membrane disruption analysis by measuring DNA and protein leakage. Additionally, we investigated membrane permeability dysfunction by staining with a fluorescence dye. We also focused on biofilm eradication, an important mechanism for microbial drug resistance. Morphological changes were further investigated under SEM. However, we acknowledge that alternative mechanisms could exist, and we considered exploring options such as resistance gene profiling and metabolic pathway analysis to rule out other possibilities. Future studies may further investigate these potential mechanisms to gain a more comprehensive understanding of EGCG's actions.
Regarding your comment, we have added the mechanism of ampicillin in the Introduction section. Please refer to line 72-74.
Revised version
“The mechanisms of action of ampicillin are well known and involve interference with cell wall synthesis through attachment to penicillin-binding proteins (PBPs), inhibition of peptidoglycan synthesis, and inactivation of inhibitors of autolytic enzymes.”
- Could you please clarify if there are any contradictions between the both antibiotics? It would be helpful to confirm their compatibility.
Response: Thank you for pointing that out. Yes, in this study, we found that one combination treatment of EGCG and ampicillin resulted in antagonism in strain E31, as indicated by an FICI of 742.19, shown in Table 2. This might be attributed to the fact that the E31 strain has a very high sensitivity to EGCG (0.002 mg/mL) on its own. However, future clinical trials may help confirm their compatibility.
Regards for your suggestion, we add more sentence in discussion. Please refer to line 336-343.
Revised version
“Although the MDR E. coli strain E31, which is resistant to six drugs (AMP, AML, KF, CTX, SXT, and SAM), exhibited good inhibitory activity with EGCG (2 µg/mL), the combi-nation treatment of EGCG and ampicillin resulted in antagonism. This antagonism may be attributed to the strain's high sensitivity to both agents. EGCG is known to interact with various biological molecules due to its phenolic structure, which enables it to bind to proteins, enzymes, and even antibiotics. This interaction could prevent these antibiotics from reaching their target sites or alter their functional efficacy, thereby demonstrating an antagonistic effect [35].”
Please see the attachment.

Round 2
Reviewer 1 Report
Comments and Suggestions for Authors
No comment
Author Response
Dear Academic Editor,
Thank you for highlighting this important point. We greatly appreciate the opportunity to address this concern.
You are correct that the growth curve of MDR E. coli in our results appears unusual. After further review and extensive efforts, we found that E. coli typically grows to approximately 10 log10 CFU/mL within 24 hours, as supported by the following references. However, our results indicated a value of up to 12 log10 CFU/mL.
To clarify, we performed a time-kill assay in triplicate independent studies (n=3), as documented in the raw data. After 24 hours of incubation, E. coli propagated to 11.23, 11.90, and 12.84 log10 CFU/mL, resulting in an average of 11.99 log10 CFU/mL with a standard deviation of 0.8.
For the negative control, we ensured that proper experimental conditions were maintained throughout the study. Thank you once again for your consideration.
Please see the attachment.
